# Design and Implementation of a Quadruped Amphibious Robot Using Duck Feet

**Saad Bin Abul Kashem** [1,*], **Shariq Jawed** [2] , **Jubaer Ahmed** [2] **and Uvais Qidwai** [3]

[1]    Faculty of Robotics and Advanced Computing, Qatar Armed Forces—Academic Bridge Program, Qatar Foundation, 24404 Doha, Qatar

[2]    Faculty of Engineering, Computing and Science, Swinburne University of Technology, 93350 Sarawak, Malaysia

[3]    Faculty of Computer Engineering Signal and Image Processing Qatar University, 24404 Doha, Qatar

*    Correspondence: skashem@qf.org.qa

**Abstract:** Roaming complexity in terrains and unexpected environments pose significant difficulties in robotic exploration of an area. In a broader sense, robots have to face two common tasks during exploration, namely, walking on the drylands and swimming through the water. This research aims to design and develop an amphibious robot, which incorporates a webbed duck feet design to walk on different terrains, swim in the water, and tackle obstructions on its way. The designed robot is compact, easy to use, and also has the abilities to work autonomously. Such a mechanism is implemented by designing a novel robotic webbed foot consisting of two hinged plates. Because of the design, the webbed feet are able to open and close with the help of water pressure. Klann linkages have been used to convert rotational motion to walking and swimming for the animal's gait. Because of its amphibian nature, the designed robot can be used for exploring tight caves, closed spaces, and moving on uneven challenging terrains such as sand, mud, or water. It is envisaged that the proposed design will be appreciated in the industry to design amphibious robots in the near future.

**Keywords:** amphibious robot; duck feet; quadruped; Klann linkage; webbed feet

## 1. Introduction

Nature-inspired robots are defining new applications as well as reviving previously abandoned explorative missions. Such exploration tasks are widely done by robots in areas where human presence is risky or impossible such as tight caves, deep oceans, or a new planet with unknown, rough terrain.

During exploration in an unknown environment, robots can experience different terrains, obstructions, and water. The robot should be capable of moving on all such terrains effectively and continue its task. Ducks are one of a variety of animals that have webbed feet. They move through the water by paddling their feet back and forth. However, this alone does not justify their efficiency when moving through the water. By observing the duck's anatomy, it can be seen that the duck has webbed feet where toes are attached by folds of skin. The objective of this research is to design and develop a prototype that will mimic the webbed feet of the duck and, consequently, is capable of roaming in diversified terrains with acceptable efficiency and effectiveness.

The presented model and design of the amphibious robot has been inspired by the working principle of duck feet. The propulsion generated through this foot system has been observed to be better and more controlled than the initial design and prototype of designed duck feet robot reported in [1]. The direction of movement can be managed, and the speed can be controlled using a larger contact area with water. Since ducks are able to roam on both land and water, their feet movement during walking and swimming is adopted in designing the robot's four legs. Klann linkage is used to

mimic such duck foot motion, and the control mechanism of the feet is driven by DC motors and DC servo motors, which are governed by an Arduino microcontroller. The robot is capable of sensing the presence of water through conductive sensors and detects obstacles using ultrasonic sensors while walking. Because of its amphibian nature and other features in movement, the robot is capable of traversing diversified terrains.

## 2. Literature Review

There are several commercially available amphibious robots that use different methods to move around on land and in water. Many robots such as quadruped robots, snake robots, and bipedal robots have been designed with inspiration from animal morphologies. Kashem et al. [1] and Dai et al. [2] observed the duck's movement underwater and found that the foot movement could be divided into two phases: stroking forward and backward phases. The backward stroking motion drives the duck body through the reacting force from the water (the duck feet fully open to maximize the contact area with water during back stroking). In the forward motion, the feet fully contract to minimize the contact area with water. Dai et al. [2] has also designed a structure of an underwater vehicle with a biomimetic propulsion mechanism. It included the body, steering engines, and propulsion mechanisms. Initially, the fins at the propulsion mechanism would be fully closed. With the swinging of the shaft, it would gradually open due to force from the water. This will give the fins enough thrust to move forward in the water. In the second phase, the fins contract due to the pressure of water from the opposite side, reducing the contact area with it. With the shaft's swinging movement, the vehicle has enough force to move forward. This design is limited because it can only move in water, and it is not functional on land as a terrestrial robot. Ijspeert et al. [3] has conducted an extensive review of locomotion control in animals and robots. Several amphibious robots have been designed and tested in recent years. Amoeba II has been designed by Li et al. [4], which is a transformable amphibious robot. The actuation system of this robot contains four main elements. Each element comprises a water-jet propeller, two servo motors, and a stainless-steel stand. Water-jet propellers actuate the robot when underwater. This design has its limitations, as it uses different mechanisms for movement in land and water. FroBot et al. [5] is a novel, amphibious robot that consists of a dual-swing-leg propulsion mechanism. Yu et al. [6] developed a bio-inspired amphibious robot, named AmphiRobot, which is capable of multimodal motion. Hyung-Jung et al. [7] designed a turtle-like robot that has a soft morphine turtle flipper using a smart soft composite (SSC) structure. A crab-like robot has been created by Chen et al. [8]. A squid-like underwater robot with two undulating side fins has been constructed and tested by Rahman et al. [9]. This robot was designed based on the median and paired fin (MPF) movement, which uses the undulating side fin for propulsion and mimics a squid. Although this robot moves slowly, it is preferred in applications where stealth is required, as in experimentations where it is important to keep the surroundings undisturbed.

Salamandra Robotica II, an advanced version of Salamandra Robotica I, is a salamander-inspired robot. This robot, designed by Crespi et al. [10], can swim in water as well as walk on land. The robot has an actuated spine and four legs that allow it to walk on the ground and anguilliform swimming in water. This robot has the disadvantages of slow movement and more chances of problems in hinge joints.

In the legged system design, numerous biological locomotors are analyzed and adapted. Animal locomotion and their walking patterns are widely researched with high interest in their complexity, flexibility, and energy efficiency. The task of designing and developing a legged robot also requires thorough optimization and a cost analysis. The design must be made by taking performance, function, and maneuverability into consideration.

As compared to wheel-based robots, legged robots have some advantages on rough terrains, as noted by Silva and Machado [11]. In terms of speed and energy consumption, the wheeled robot is far superior to the legged robot. However, many researchers are designing improved legged robots with enhanced performances. Raibert et al. [12] designed Big Dog, which has four legs and is quite

effective in performing various locomotion and logistical tasks. Lokhande and Emche [13] designed a walking robot, which follows the locomotion of a spider and utilizes the Klann linkage mechanism. Similarly, Zhang and Kimura [14] fabricated a quadruped robot, named Rush, by imitating legged animal movement for their robots. Their mechanism is complex and uses numerous sensors to help the robot walk. However, all of these robots are designed for land-based terrains only and are not amphibious in nature.

In this research, Klann linkage has been used to design the leg mechanism. Sheba et al. [15] suggested putting actuators among the connections of Klann linkages in order for it to perform better in walking mode. However, such intricate linkage is expensive and not necessary in prototype design. Thus, the fundamental Klann linkage is adopted in this project. The foot design of the robot is inspired by the biological webbed foot design of a duck.

Ducks belong to the amphibian category of animals that can travel both on land and water. They move through the water by paddling their feet back and forth. However, this alone does not justify their efficiency when moving through water. By looking at the anatomy of a duck, it can be observed that the feet of a duck are webbed, meaning the toes are connected by folds of skin. This provides a more efficient transfer of force when moving through water.

Ribak et al. [16] explained in their research that the maximum propulsive force is generated when the robots' feet are swept backward in the water.

When in water, the robot moves using drag-based swimming. The mechanism proposed by Li et al. [17] is not feasible for the proposed model since the spherical robot has two different mechanisms for locomotion. The swimming mechanisms of the robots designed by Dhull et al. [18], Dudek et al. [19], and by Liang et al. [20] were somewhat similar compared to the robot developed in this paper. Figure 1 shows the Aqua and Aquapod robot. The Aquapod robot has GPS (Global Positioning Center) to navigate. This Aquapod robot has also used PTC (Push to Connect) Fittings. The basic difference is that they utilize rotational arms to move their robot through water and land. Whereas, the presented robot in this paper has been designed using a proper legged mechanism, which can support swimming and suitable walking phenomena. The initial design and prototype of the designed duck feet robot are reported in [1], and some modifications were done in [21–26]. In this manuscript, the final design, layout of the electrical circuit, the final prototype with dimensions, and testing results have been provided. The propulsion generated through the foot system is better and more controlled than the initial designs. The direction of movement can be managed, and the speed can be controlled using a larger contact area with water.

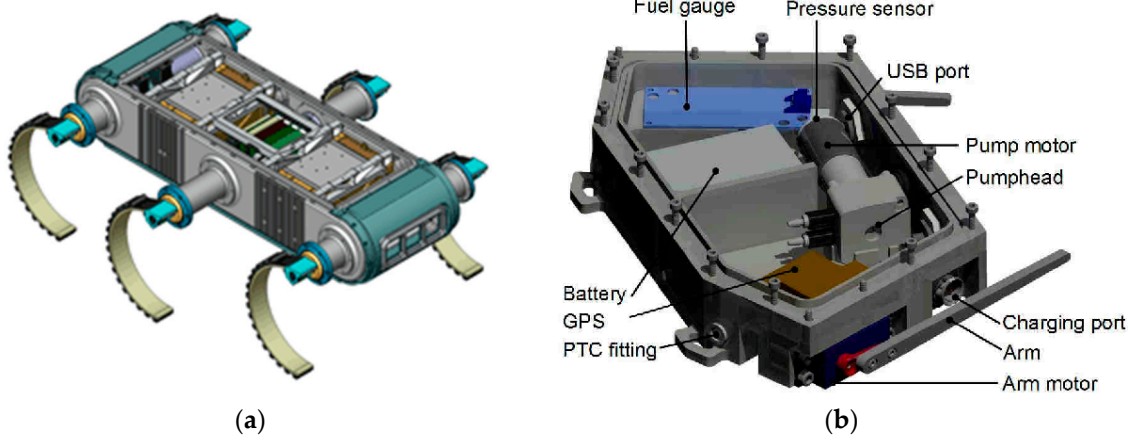

**Figure 1.** (**a**) Aqua robot and (**b**) Aquapod Robot.

## 3. Design and Methodology

### 3.1. Mechanical Design

Design of the duck feet was completed in SolidWorks software environment first. To check the validity of the design, simulations were conducted. The linkages and the whole body were simultaneously simulated. The feet were engineered to have a webbed formation that mimicked duck feet, as shown in Figure 2. The figure shows the design from various viewpoints and is annotated with the dimensions.

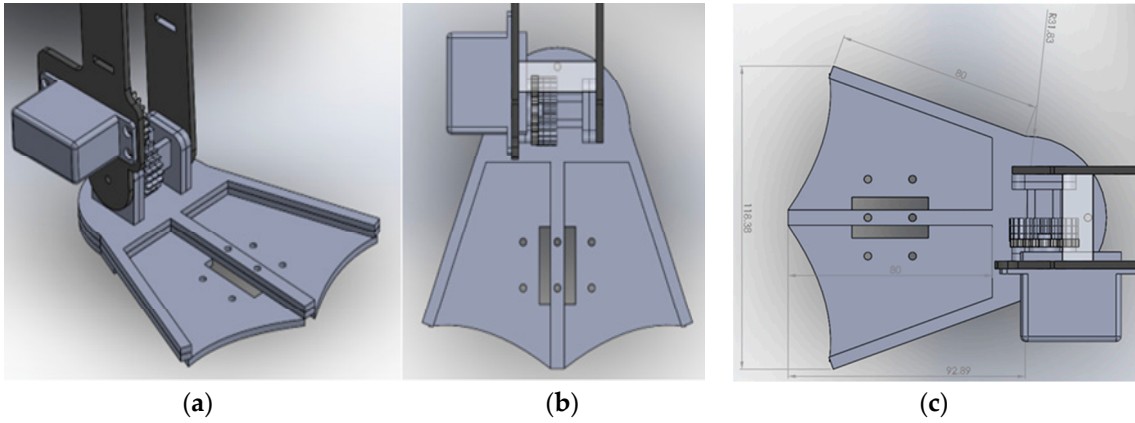

| (a) | (b) | (c) |

**Figure 2.** Foot design. (**a**) Isometric view, (**b**) top view, and (**c**) top view with dimensions.

By the aid of two hinges, the two flaps were linked to the bottom of the foot. For walking and swimming purposes, the foot base was made such that it provided a large contact area to provide stability while standing and to push forward in the water. The feet were connected to the respective Klan linkage. The angle of the foot can be changed by a high-torque servo motor, as shown in Figure 3. The foot angles in walking and swimming mode were 0° and 80°, and the angle remained fixed during foot movement.

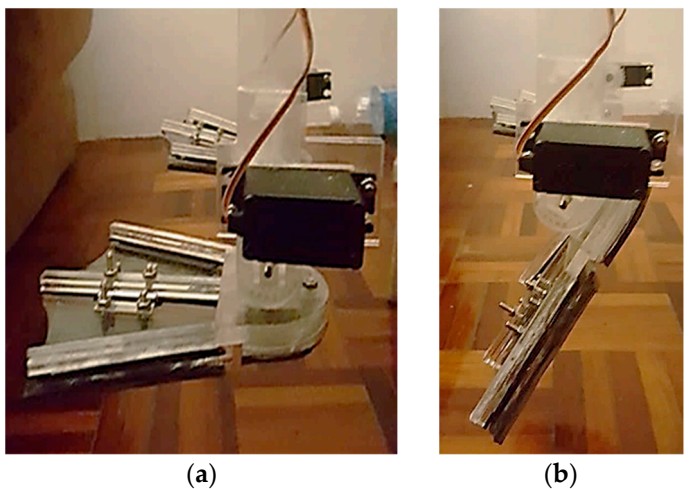

| (a) | (b) |

**Figure 3.** Foot orientation during (**a**) walking mode and (**b**) swimming mode.

This design of the feet allows for the duck to walk without falling over during walking. To achieve stability of the structure, the robot was converted to a quadruped by adding four identical feet to the robot's body. The feet were set in such a way that, at any given time, two feet always remained in contact with the ground. During swimming in the water, four feet can go forward and backward at the same time to maximize the speed. In swim mode, both motors will go back and forward at the same

time since the robot will move faster than if both motors went back and forward at different times. This has been observed during practical tests. No data were recorded, as the main focus was the leg design. The flaps of the foot close automatically when pushing back to maximize the contact area with water, as shown in Figure 4a. This helps to push the maximum amount of water in the backward direction, which helps the robot move forward. On the other hand, when the feet move forward, the flaps open automatically to minimize the force required by the motors since the feet obstruct less water, as presented in Figure 4b.

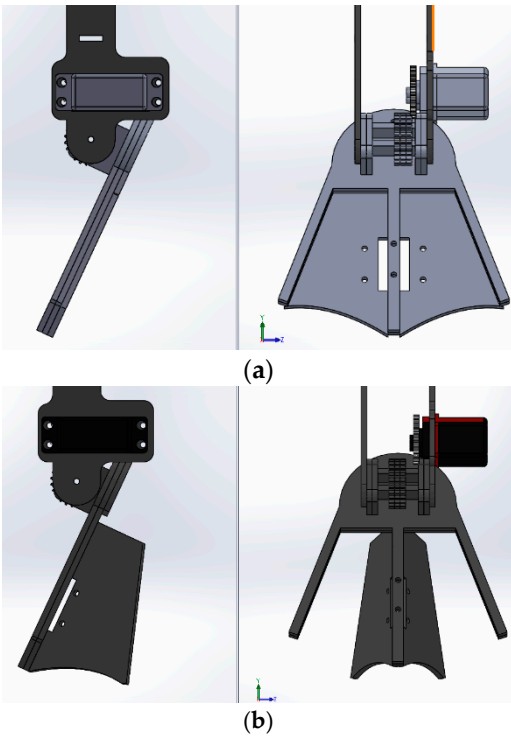

(a)

(b)

**Figure 4.** Duck foot position during (**a**) backward motion and (**b**) forward motion.

Figure 5 shows the interconnected joint movements in the overall foot/leg assembly. This is shown with the help of the trace path of the Klan linkage and shows various joint elements in terms of their displaced positions with respect to the various positions during the walking/swimming movements. The actual design dimensions are also illustrated in the figure. The vertical distance covered by this linkage is approximately 3 cm, whereas the horizontal distance is approximately 7.5 cm, which has been shown graphically in Figure 6.

The full model's isometric view and top view are shown in Figure 7a, b, respectively. The drag force is the force opposing the movement of the robot in water. The force is applied on the feet by the water and is the force the robot will face, which will make it move forward.

The drag force applied on the feet was calculated using Equation (1).

$$F_D = \frac{1}{2}\rho v^2 C_D A,$$
(1)

where

- $\rho$ is the density of the fluid;
- v is the velocity of the object;
- $C_D$ is the drag coefficient; and
- A is the area of the object.

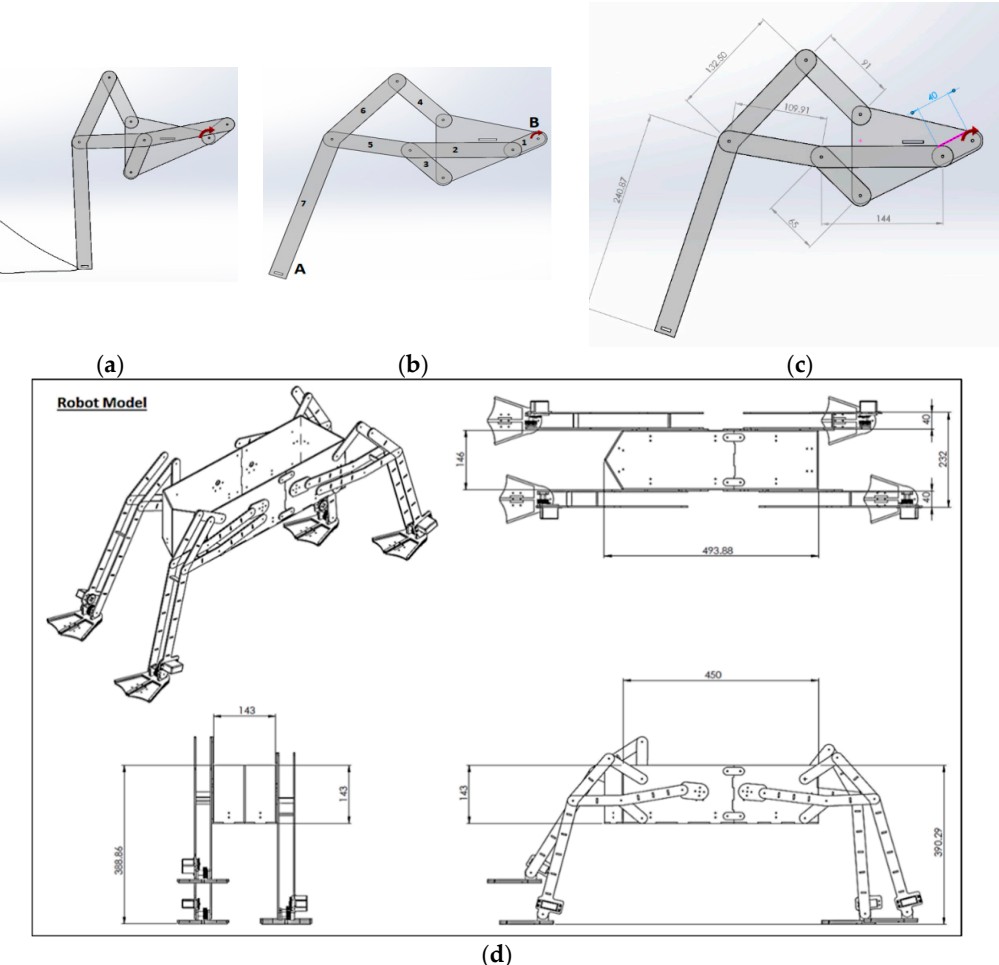

**Figure 5.** Tracing the path of the Klan linkage when the foot is in the (**a**) closed position and (**b**) extended position. (**c**) Linkage with dimensions. (**d**) The actual design dimensions of the robot.

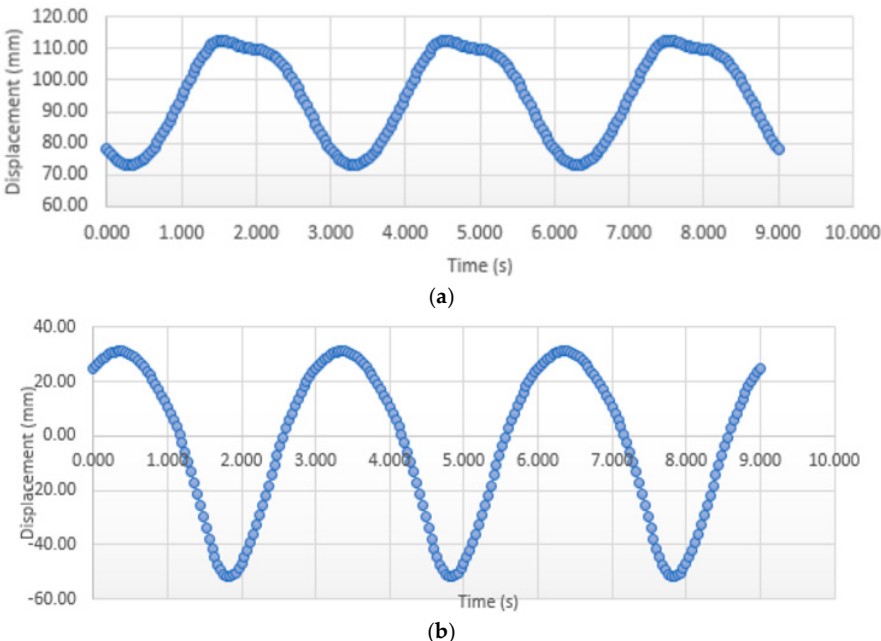

**Figure 6.** (**a**) Vertical displacement and (**b**) horizontal displacement.

Mybotic DC gear motor JGB37 was used in this model. Based on the motor specifications, the velocity of the feet was computed as 0.03125 m/s. The density of water is 1000 kg/m$^3$, and the drag coefficient was calculated appropriately.

The area was computed with the aid of Solidworks, as shown in Figure 8.

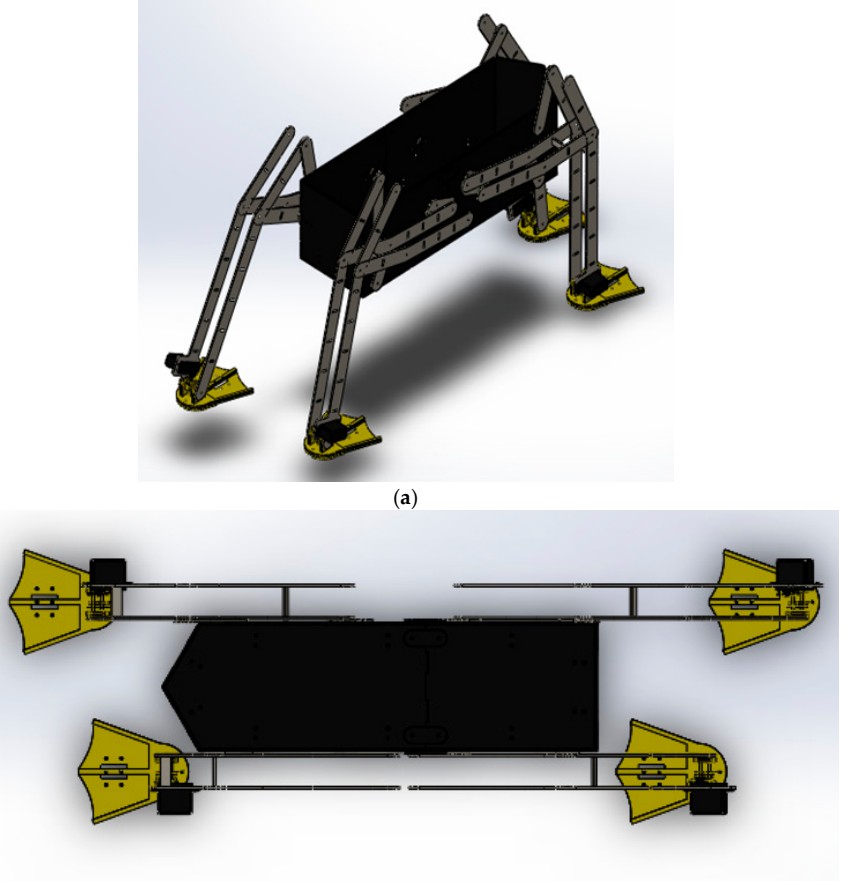

(a)

(b)

**Figure 7.** (**a**) Full model isometric view and (**b**) full model top view.

Mass properties of Feet Bottom part
   Configuration: Default
   Coordinate system: -- default --

Density = 0.00 grams per cubic millimeter

Mass = 11.89 grams

Volume = 9908.87 cubic millimeters

Surface area = 8948.50 square millimeters

Center of mass: ( millimeters )
    X = 0.00
    Y = 1.50
    Z = 12.47

(a)

Mass properties of FLAP
   Configuration: Default
   Coordinate system: -- default --

Density = 0.00 grams per cubic millimeter

Mass = 9.82 grams

Volume = 8182.85 cubic millimeters

Surface area = 6246.74 square millimeters

Center of mass: ( millimeters )
    X = 21.47
    Y = 1.50
    Z = 37.51

(b)

**Figure 8.** Surface area of the (**a**) bottom of the feet and (**b**) flaps.

As there are two flaps and a foot bottom for each foot, which pushes the water at any given moment, the total theoretical surface area is

$$0.00895 \text{ m}^2 + 0.00625 \text{ m}^2 + 0.00625 \text{ m}^2 = 0.02145 \text{ m}^2;$$

$$0.02145 \text{ m}^2 \times 2 = 0.0429 \text{ m}^2.$$

The drag force becomes

$$F_D = \frac{1}{2}(1000)(0.03125)^2(1)(0.0429) = 0.0209 \text{ N}. \tag{2}$$

So, the force being applied to the robot from water is 0.01341 N.

Now, the acceleration of the robot was calculated as:

$$a = \frac{F}{m} = \frac{0.0209}{1.89 \text{ kg}} = 11.06 \times 10^{-3} \text{ m/s}^2. \tag{3}$$

The mass was taken from Solidworks, as in Figure 9. The robot will move with an acceleration of 0.01106 m/s$^2$.

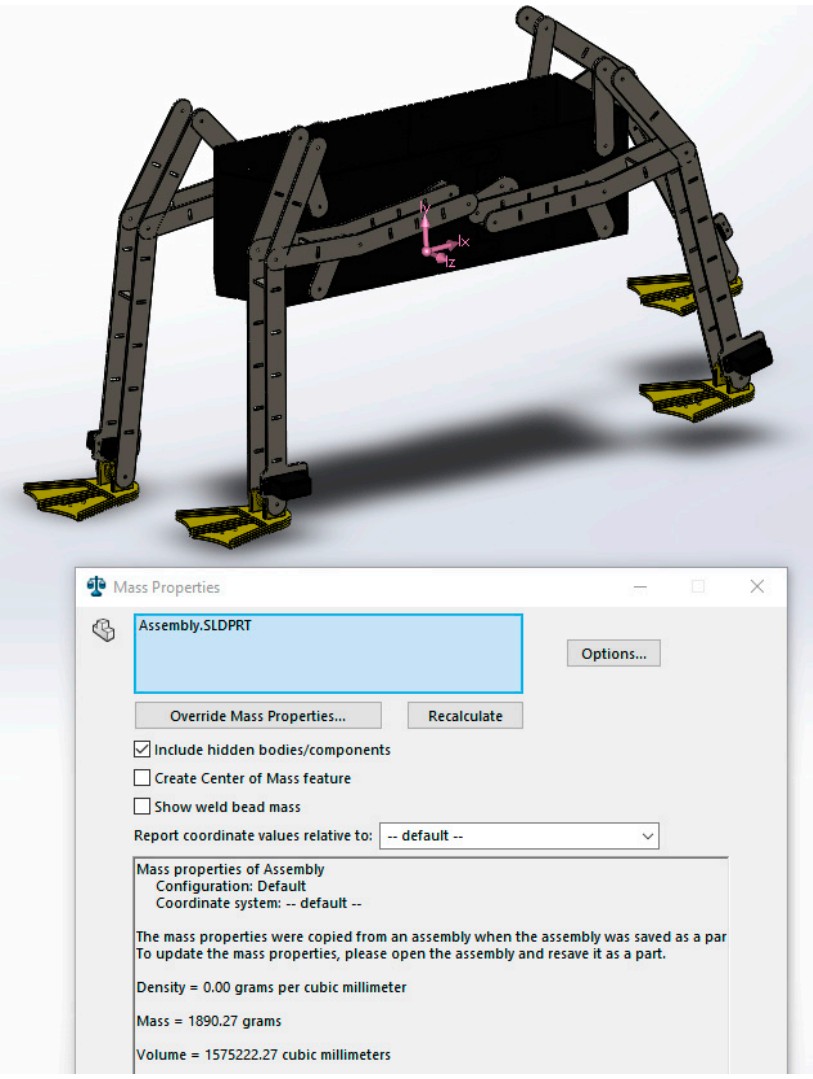

**Figure 9.** Mass properties of the robot.

*3.2. Electrical Design*

To design appropriate functions of the robot, several combinations of sensors and actuators have been designed. An Arduino Mega microcontroller has been used to control these sensing/actuating components. Selection of the actuators is specifically important since the whole body of the robot stands on these. Figure 9 shows various components used in the design.

Waterproof EMAX ES3005 servo motors were used to design the robot because of their high torque and low-speed capabilities. They have a speed of 0.14 s/60° and a torque of 12 kg/cm. These servo motors perform the main job of changing the mode from walking to swimming and vice versa. In addition to that, servo motors provide the forward motions. Four servos were used in a coordinated manner, one on each foot.

Apart from servo motors, a Mybotic DC gear motor JGB37 was used to control the linkage of the forward movement. It has a speed of 30 rpm and a rated load torque of 25 kg/cm. Every foot was affixed with one DC motor, which was paired with a Hall Effect sensor (built-in with motor) for feedback positions, to control the leg movement and to avoid any desynchronization.

A JSN-SR04T waterproof ultrasonic sensor (range of up to 4.5 m) and a water sensor were used in this robot, as shown in Figure 10a,b, respectively. The ultrasonic sensor was used to detect any obstacle in front so that the robot could avoid it. In addition to that, it measures the water depth to perform switching from walking to swimming. The water sensor was used to detect the presence of water. Whenever the presence of water was detected with sufficient depth for swimming, the robot instantaneously switches from walking to swimming mode.

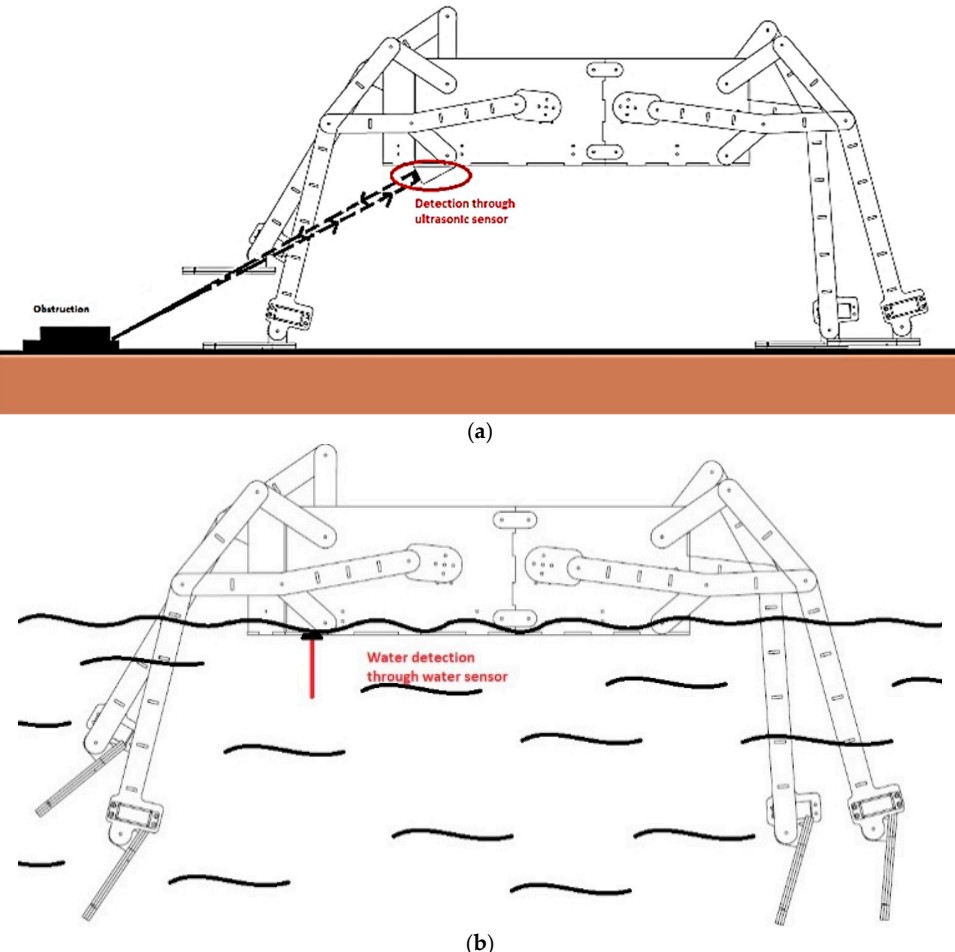

(a)

(b)

**Figure 10.** (**a**) Obstruction detection with the ultrasonic sensor. (**b**) Water sensor to sense the presence of water.

For the power supply, compact and efficient 18650 lithium-ion batteries were used, which have a high capacity and are rechargeable. These batteries have a rated capacity of 3800 mAh and provide 3.7 V. Such batteries were highly suitable for this project because of their compact nature, and, hence, a total of 6 batteries were used (two parallel banks of three batteries connected in series) in order to get a combined voltage of 11.1 V with enhanced current. Figure 11 illustrates the internal connections of the electrical components detailed above. The same block diagram is converted into a circuit layout in Figure 12. Finally, Figure 13 depicts the complete design of the operational circuit board.

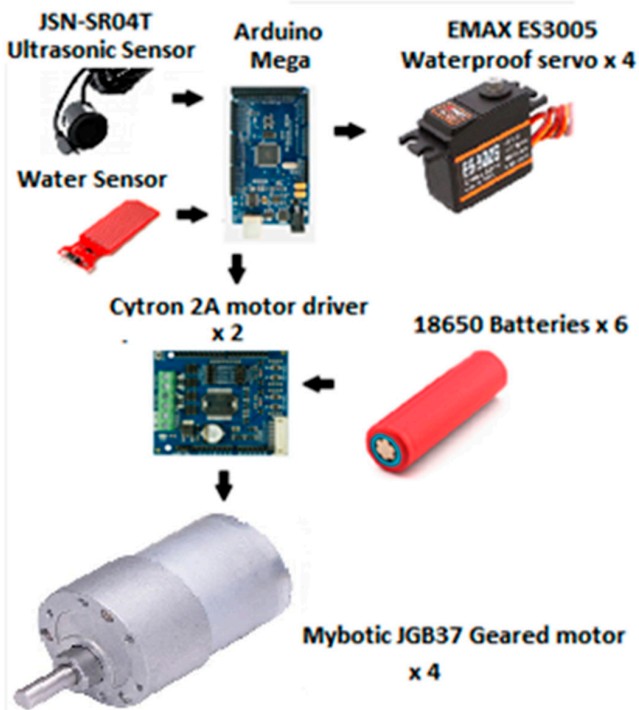

**Figure 11.** Electrical components used in the design.

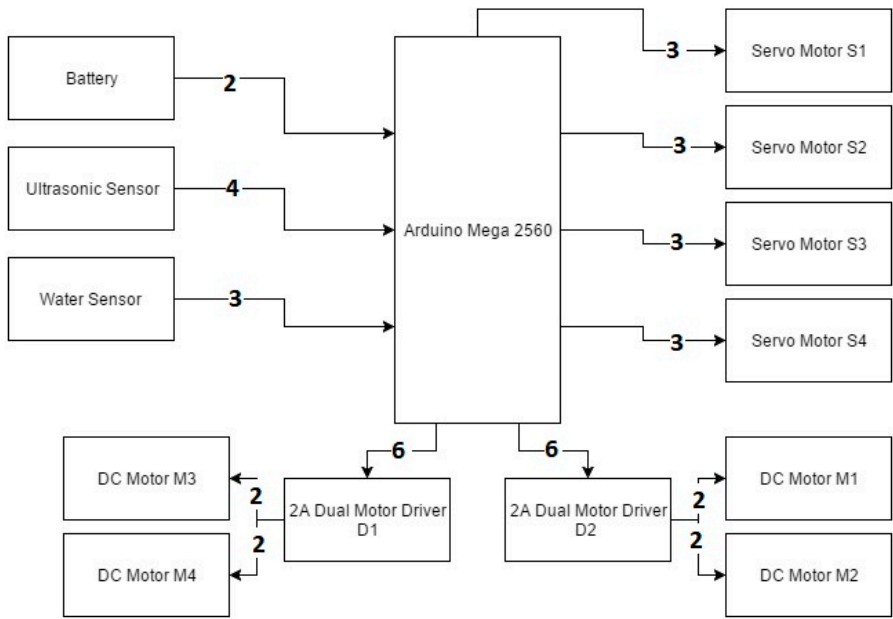

**Figure 12.** Block diagram showing the electrical connection layout.

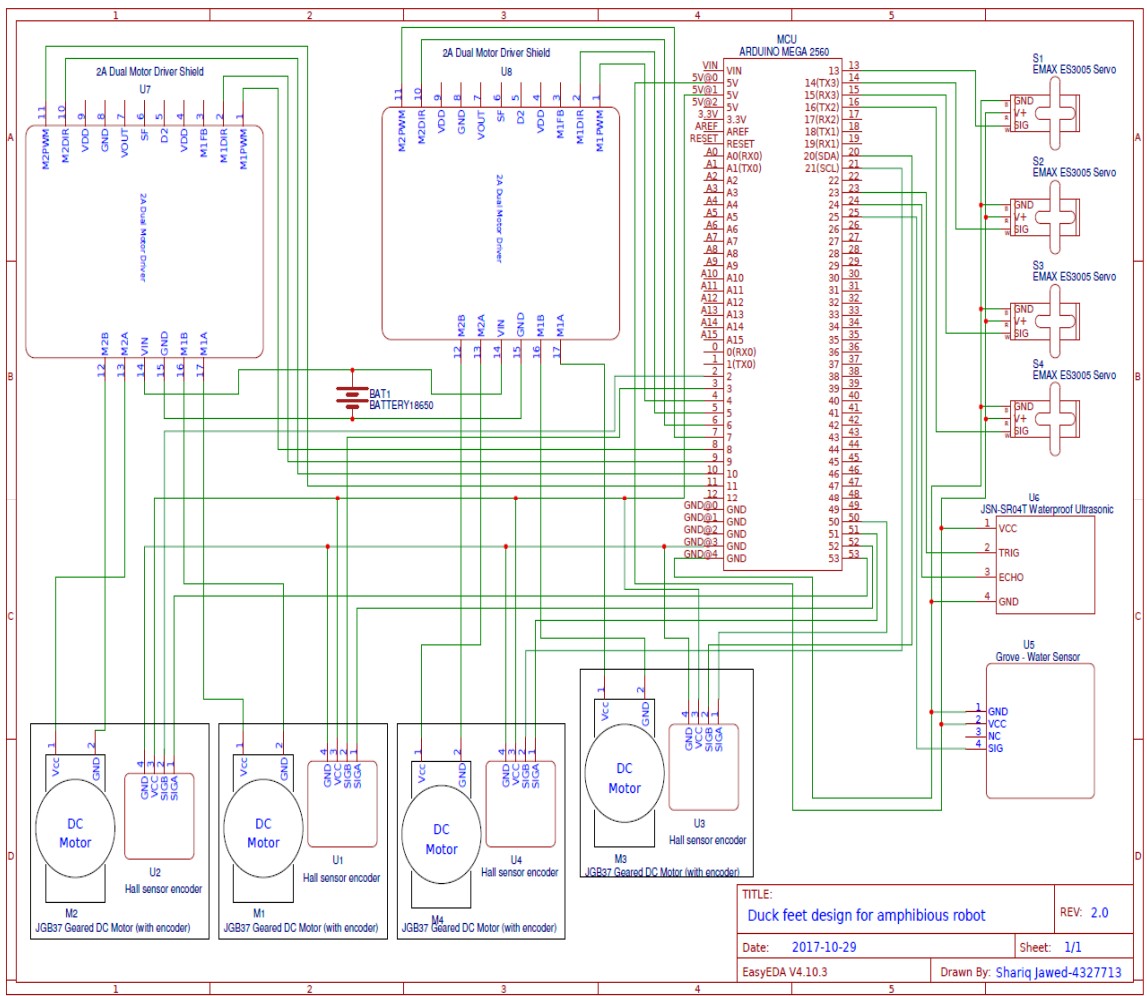

**Figure 13.** Detailed circuit diagram of the overall system.

*3.3. Program Flow*

The completed robot was first tested for basic functionalities and maneuvers. The flowchart for the program that performed these tests is shown in Figure 14. The robot was activated and took the initial position once button A on the remote was pressed. Once button B on the remote was turned ON, the water sensors checked the presence of water and activated the swimming mode if the water was present, otherwise it activated the walking mode. In the next step, the ultrasonic sensor detected any obstacles within the measurement range in front of the robot as it started to move. With no obstacles detected, the robot will check the status of button B first. If button B is ON, then the robot moves forward for 5 s and runs the loop again. In the case where an obstacle is detected in the path, the robot will turn right for 2 s and go straight for 5 s then turn left for 2 s. Once the turns are done, the robot will continue to go forward for 5 s and repeat the loop as long as button B on the remote is in the ON state. The robot will take the final position and turn off when button B is in the OFF state. The B button on the remote control was used as an emergency switch to avoid any hazardous situations. For safe handling of an autonomous robot, an emergency shut down switch is important. To turn right during swimming mode, the left motors rotate with maximum power, and the right motors get zero power. On the occasion of turning right during the walking mode, the left motors rotate with maximum power, and right motors get 60% power to keep the robot stable.

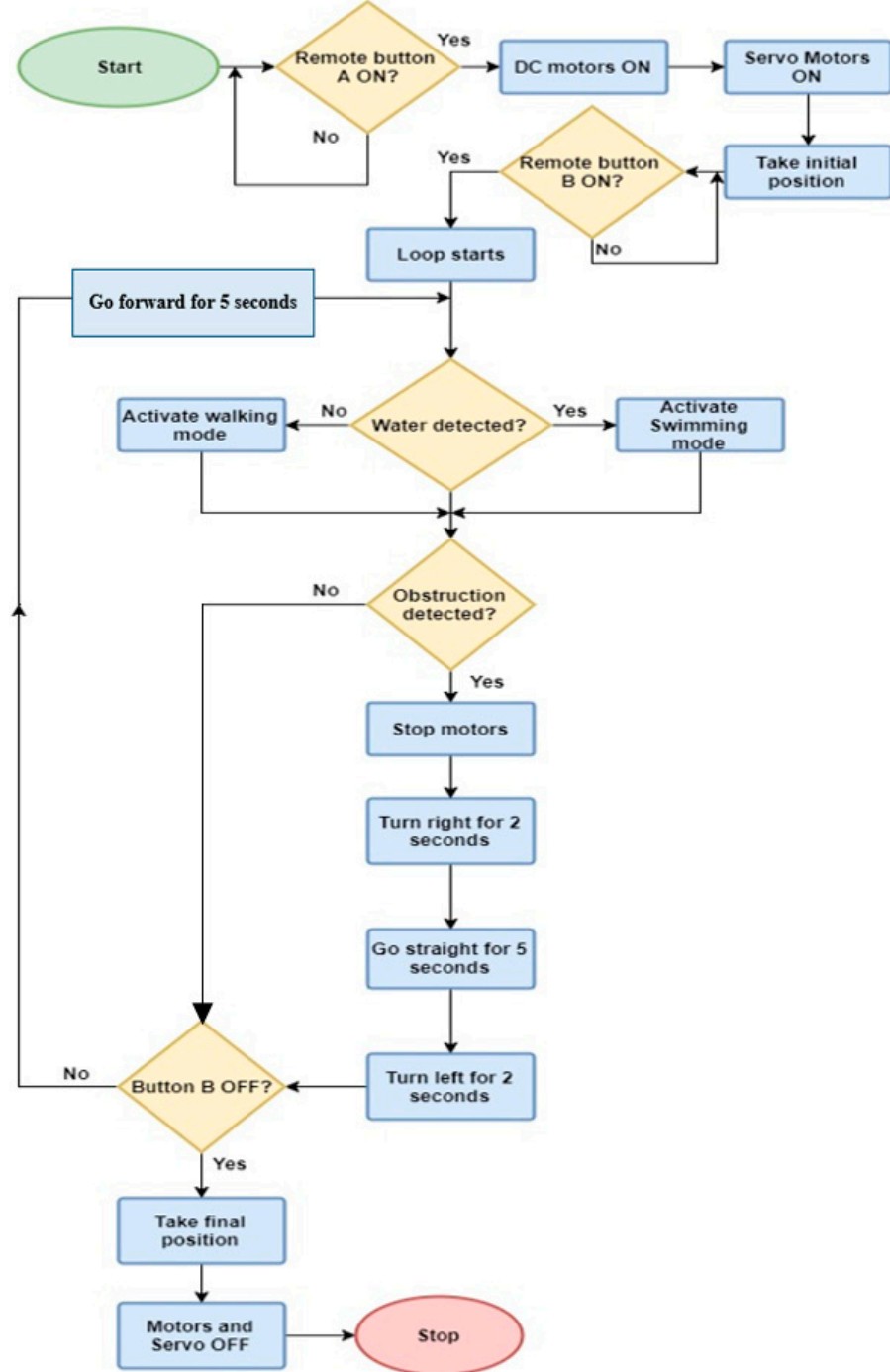

**Figure 14.** Program flow.

## 4. Discussion and Results

The prototype was designed considering the balancing issue of the robot. The weight distribution of the robot was carefully done, and batteries were installed on the upper deck of the robot's body to allow easy recharge or swap. The completed prototype's isometric and top views are shown in Figure 15.

The robot was capable of moving approximately 7.5 cm horizontally per rotation of the motor. The no-load speed and full load speed were 30 rpm and around 25 rpm, respectively. As a full 360° rotation with load took 2.4 s by the motor, the velocity of each leg becomes 0.03125 m/s. In one minute, the robot was able to travel about 1.9 m with continuous velocity.

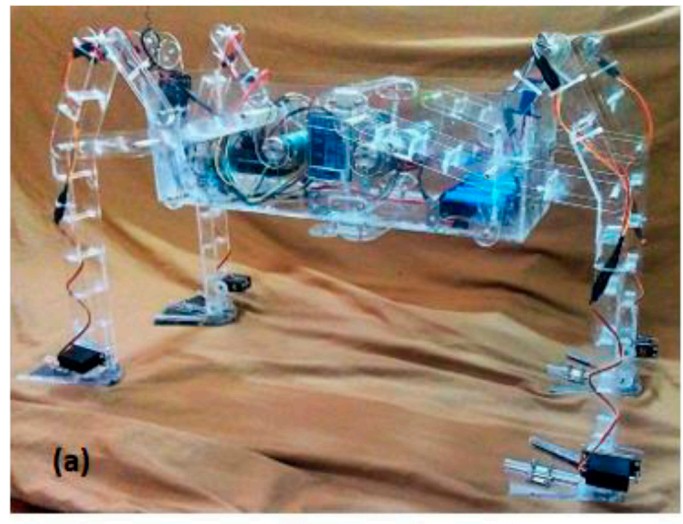

(a)

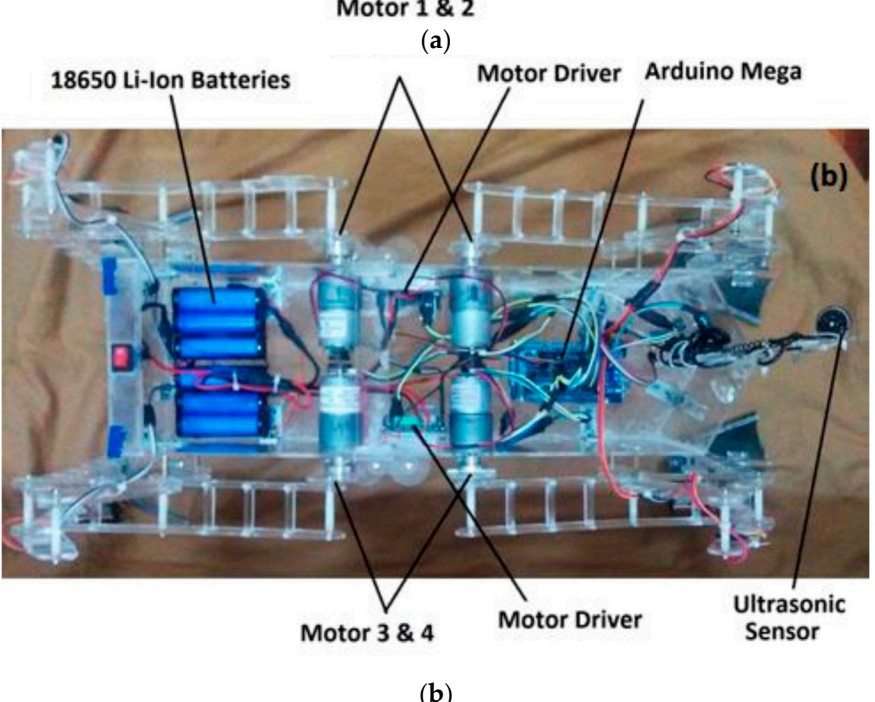

(b)

**Figure 15.** Completed prototype of the robot, (**a**) Isometric layout of the prototype and (**b**) top view of the design.

When the water sensor, attached under the body of the robot, detects the presence of water, it provides the necessary signal to the controller. The controller sends the activation signal to the actuator attached to the legs to rotate and change the leg positions from walking to swimming mode. By the time the water reaches the water sensor (body level), the robot starts floating. This makes it safe to move the legs in swimming mode without hitting the ground.

The robot also has the option to detect an obstacle (shown in Figure 16) using an ultrasonic sensor, which was attached to the front of the body (shown in Figure 10a). According to the design, the robot will automatically stop for a second when it detects an obstacle 20 cm away. Then, the robot will turn right for 2 s and go straight for 5 s. Then, it turns left for 2 s, thus avoiding the obstacle. Once the turns are done, the robot will continue to go forward for 5 s and repeat the loop as long as remote button B is in the ON state.

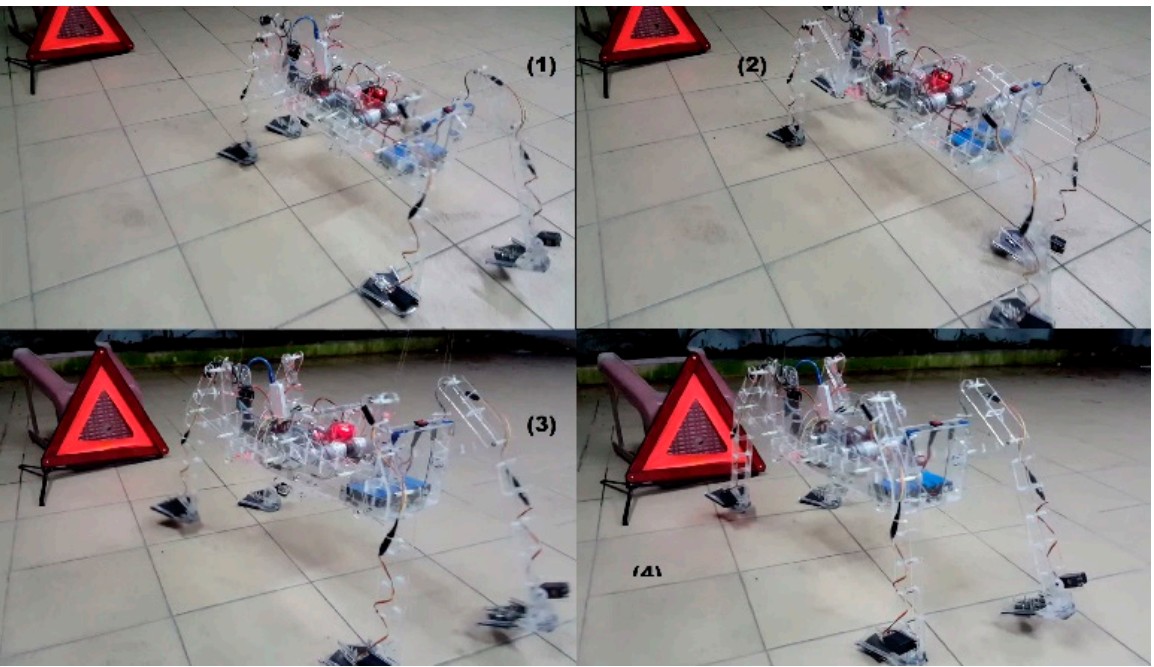

**Figure 16.** Obstruction detection by the prototype.

## 4.1. Walking Test

The walking functionality of the robot was tested on land, and synchronization of leg movement was observed. It was noted that by using the Hall effect sensors to control the speed of the robot, the legs were synchronized and did not deviate from their positions with time. The robot walking is shown in Figure 17, while the motor coordination is shown in Figure 18. The video of these tests can be seen at https://youtu.be/du9R-pkJYzc.

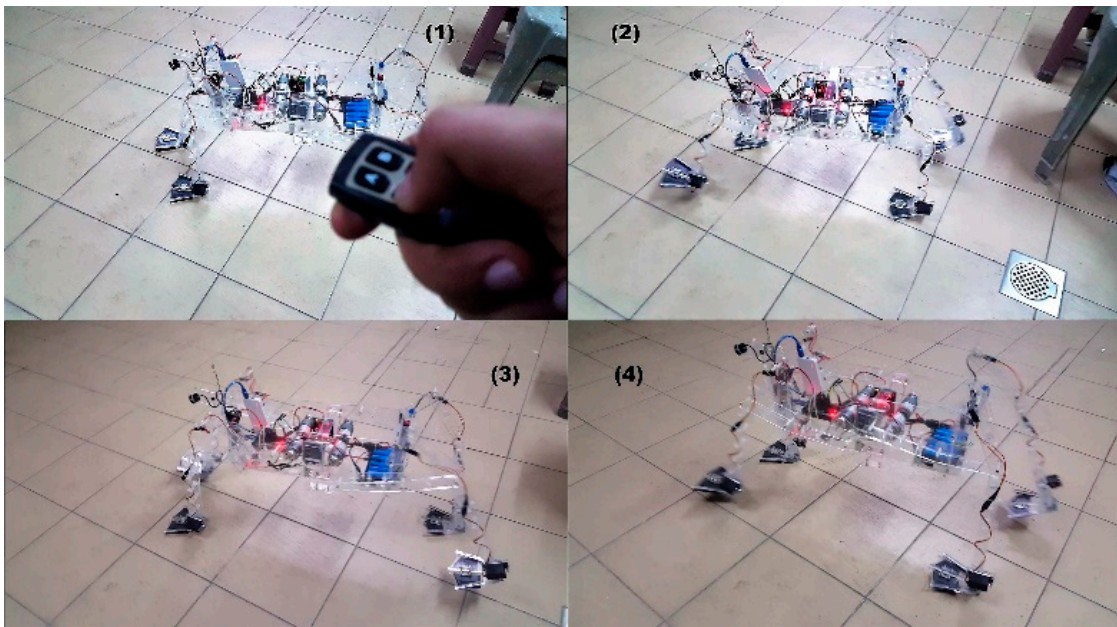

**Figure 17.** Walking test of the prototype.

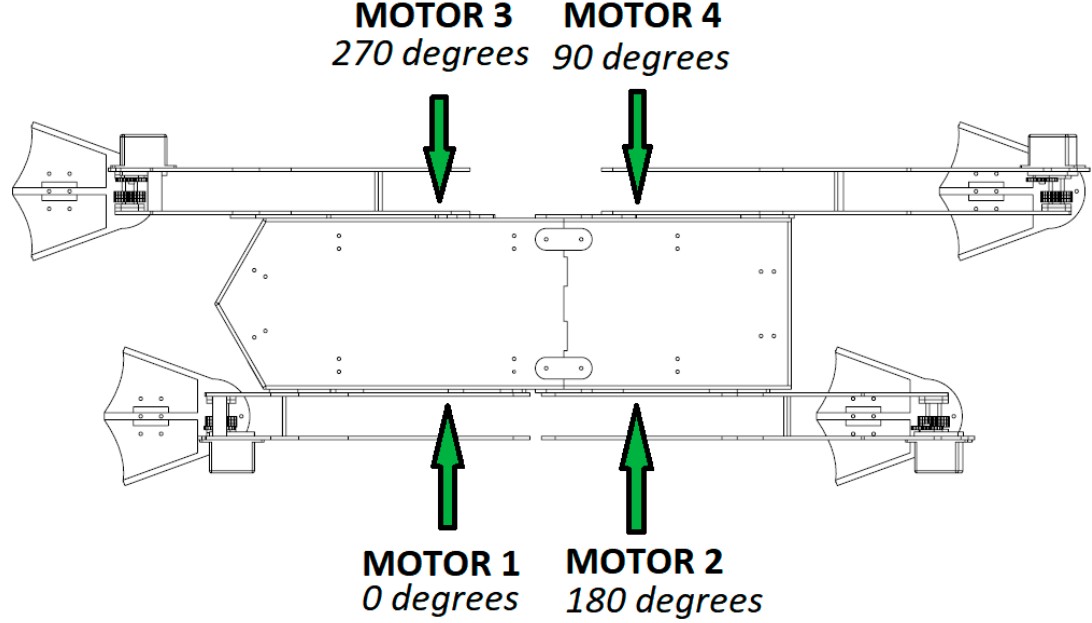

**Figure 18.** Motor coordination in the robot.

*4.2. Swimming Test*

The robot was tested in water to observe its movement and speed. It moved slowly compared to the walking mode. To improve the speed while in the water, the foot surface area can be increased to surge-up the force exerted by them. Figure 19 shows a swimming test, and Figure 20 depicts opening and closing of the flaps while swimming. The video of swimming tests can be seen at https://youtu.be/KjaWv5tcZyM.

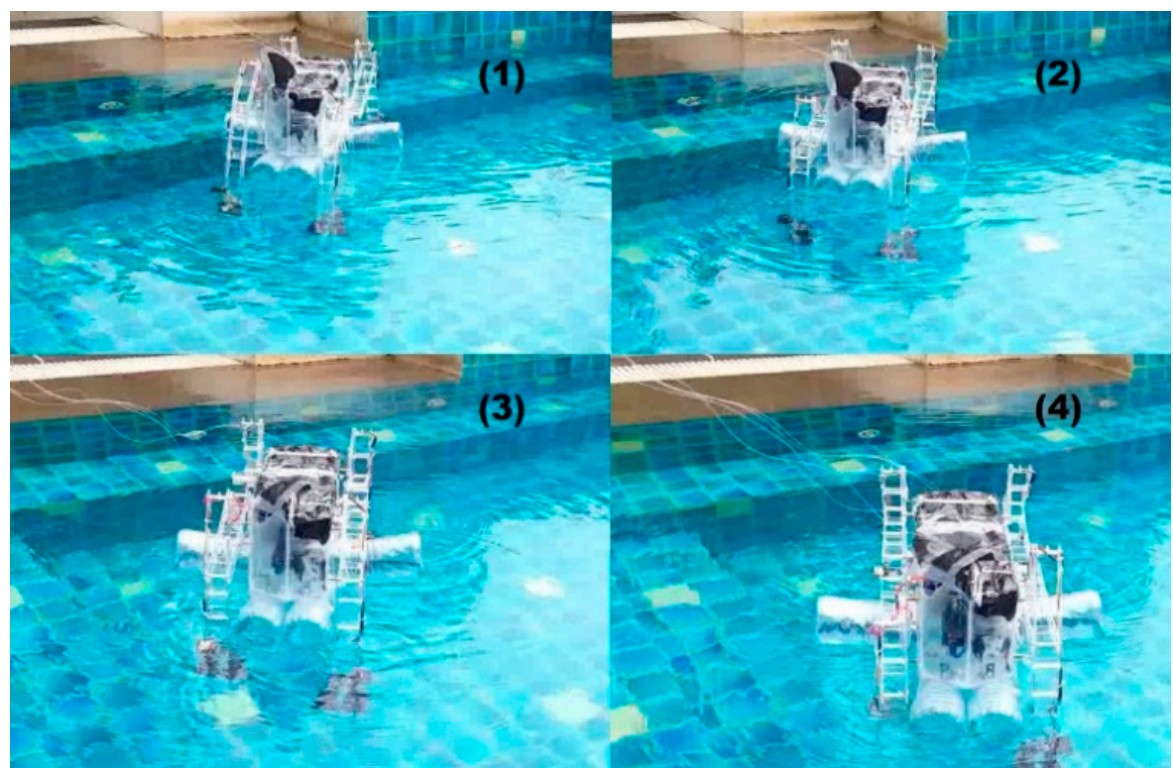

**Figure 19.** Swimming test of the prototype.

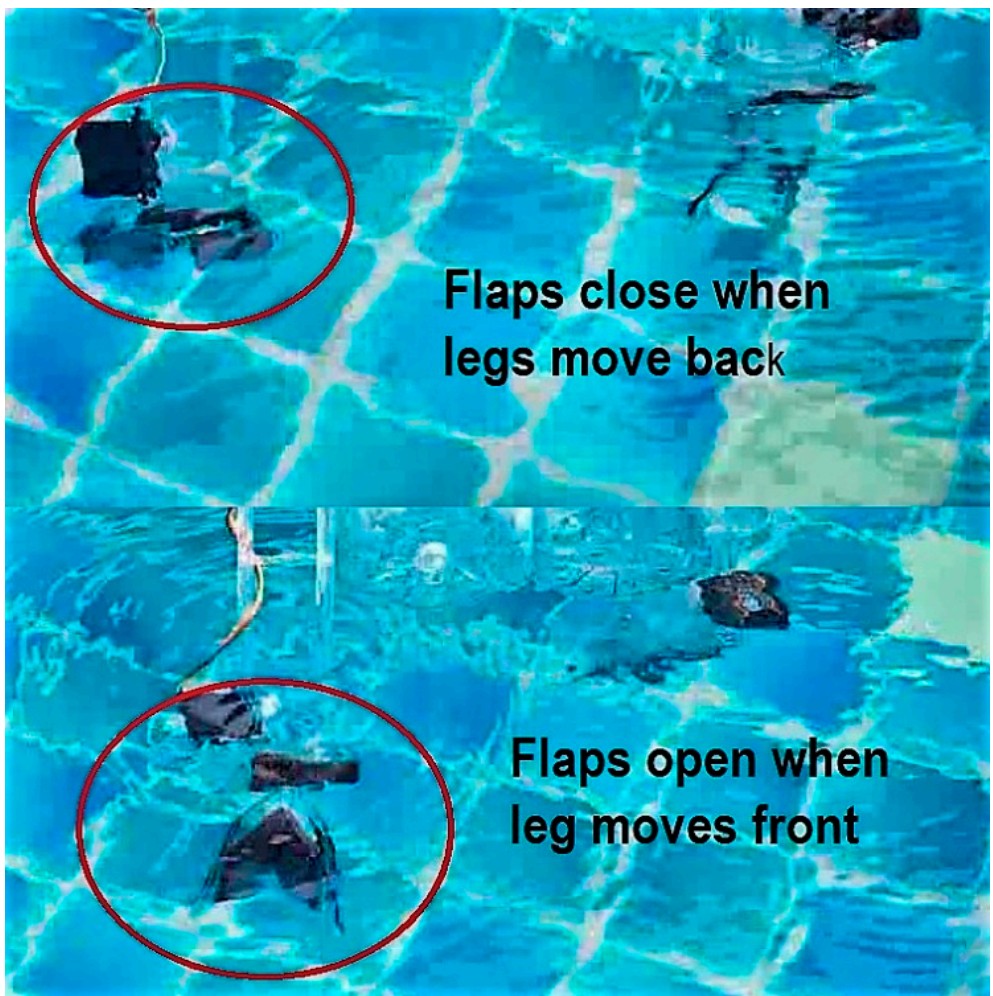

**Figure 20.** Flap opening and closing while swimming.

## 5. Conclusions and Future Work

In this paper, the model and design of an amphibious robot are presented. The proposed design mimics the operational principles of duck feet to move on land and in the water. The movement of the duck feet is critically analyzed, and duck feet were replicated by using improved Klann linkages. The body of the robot was made proportionately so that it could be carried by four duck feet and was able to carry all control circuitry. Finally, the designed prototype was tested on land and in water, where the robot successfully walked and swam, respectively. It should be noted that the performance of this robot is not up to the efficiency of wheeled robots on smooth terrains or propulsion-based robots in water. Rather, it is expected that this work will be recognized as a unique idea that combines walking and swimming under a unified mechanism. Nevertheless, the proposed idea can be improved further by choosing lighter materials instead of acrylic. Moreover, an additional sensor can be used to detect obstacles on both sides of the robot rather than only detecting obstacles in front of the robot. Besides, reconfigurable Klann linkages in place of static ones will improve the mobility of the robot significantly.

**Author Contributions:** S.B.A.K. devised the project, the main conceptual ideas and proof outline. S.B.A.K. were involved in planning and supervised the work, S.J. and S.B.A.K. processed the experimental data, performed the analysis, drafted the manuscript and designed the figures. S.J. performed all the calculations. J.A. and U.Q. verified the analytical methods. All authors discussed the results and contributed to the final manuscript.

**Funding:** This research received no external funding.

**Conflicts of Interest:** There is conflict of interest.

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
