# Peer review of "Design and Implementation of a Quadruped Amphibious Robot Using Duck Feet"

_robotics, doi:10.3390/robotics8030077_

Round 1

Reviewer 1 Report

This research aims to design and develop an amphibious robot, which incorporates webbed duck feet design to walk on different terrains, swim in water and tackle obstructions on its way. It is an interesting report that has merit, but needs to be extensively revised according to the comments.

1. The abstract is quiet long, and it need to be simplified.

2. The figures are not clear, which is need to be corrected, such as Figure 1, Figure 9, and Figure 12.

3. Figure 6 need to be corrected. The format need to be unified to other figures. and in some figures, the format is realtive small.

4. English must be polished.

5. In this paper, the design of walking gaits and swimming gaits is not mentioned. However, they are relized in walking and swimming experiments. It caused the incompleteness of this paper. These parts need to supplement. What's more, there is no experimental results analyses in experiment part, such as the walking and swimming velocity, the stability of this robot while moving.

Reviewer 2 Report

The article presents the design and prototype of a bio-inspired duck robot able to walk and swim using the same mechanism thanks to its webbed feet. It is an interesting work because the robot can operate in dangerous places like tight caves and it seems to be a cheaper solution because it only uses one mechanism of propulsion.

The authors present related works that propose bio-inspired approaches such as amoeba, FroBot, AmphiRobot, turtle, squid, crab, Salamandra, spider and dog. Although some of them were cited without giving more details, the authors conclude that some of them adopted the Klann linkage and none of them can swim and walk with the same propulsion mechanism. Please, consider reviewing the authors' name and citations in this Section, Tuan Dai is not the author of articles [1] and [3] and A. J. Ijspeert is not the author of paper [2]. 

The text is Ok, and I do not find typos or critical writing problems. However, some phrases can be shorted or reorganized to avoid repetitions (Highlighted in orange on the attached PDF).

The remainder commentary is related to question and suggestion regarding the article.

In Section 3, it is expected a discussion about how the simulations were conducted. Are they conducted on SolidWorks too? Also, How? There are results and conclusion of these simulations? The simulated result significantly improves the article.

Regarding the prototype, what is the total weight, height, and length? Where is the depth sensor installed? Also, what is the feet angle in swim mode and walking? Is it a fixed value?

Although Figure 3 shows the feet position, it is not clear what is the angle. Also, the figure is never referenced in the text.

As mentioned by the authors, a discussion about robot efficiency and maneuverability is interesting in that kind of article. Although some information like robot speed and displacement by motor turns are shown, it is not clear how the robot perform a turn in the swim and on the walking mode? What is the robot maximum turning angle? About the prototype, what is its power consumption or how long does a battery pack last in constant velocity in the swim and the walking mode? There is an estimation of the maximum payload weight? The authors mention that in swim mode both motors went back and forward at the same time, it would be interesting a discussion about the decision.

Please consider improving the image quality of Figure 9. Image of JGB37 motor has some white and indistinguishable letters on the top.

Please, consider changing figures 1, 6 and 13 in order to follow the same reference style of the subfigures of figures 2,4 and 5.

Please consider improving the image quality of Figure 12. It seems it was affected by the image compression process; Maybe you can improve it exporting the diagram to a format with a minimum loss like "png" instead of "jpg" or using vectorized files such "eps".

Regarding the program flow of Figure 12, it seems the robot only stop if an obstruction is detected AND the B button is off. Only turning the B button off is not enough to make the robot stop. It may be not the desired behavior as described in the text of Section 4. Also, it seems there is a walking forward instruction missing on the diagram when no obstacles are detected.

It is not clear in Figure 18 that the flaps are closed or open. Please consider improving this figure. I believe a better picture of the open and close flaps can be extracted from the youtube video of the swim test that was not included in the article. Also, include the swim test video on the article.

I attached the PDF I worked on the review process of this article.

Author Response

Reviewer #1:

Comments and Suggestions for Authors

The article presents the design and prototype of a bio-inspired duck robot able to walk and swim using the same mechanism thanks to its webbed feet. It is an interesting work because the robot can operate in dangerous places like tight caves and it seems to be a cheaper solution because it only uses one mechanism of propulsion.

The authors present related works that propose bio-inspired approaches such as amoeba, FroBot, AmphiRobot, turtle, squid, crab, Salamandra, spider and dog. Although some of them were cited without giving more details, the authors conclude that some of them adopted the Klann linkage and none of them can swim and walk with the same propulsion mechanism. Please, consider reviewing the authors' name and citations in this Section, Tuan Dai is not the author of articles [1] and [3] and A. J. Ijspeert is not the author of paper [2].

We really thank the reviewer for pointing out this aspect. There was an error during the citation. Now we have updated the paper.

The text is Ok, and I do not find typos or critical writing problems. However, some phrases can be shorted or reorganized to avoid repetitions (Highlighted in orange on the attached PDF).

The text Highlighted in orange have been rephrased

The remainder commentary is related to question and suggestion regarding the article.

The answers have been given at the attached Pdf file (robotics-498185-review 2019 response)

In Section 3, it is expected a discussion about how the simulations were conducted. Are they conducted on SolidWorks too? Also, How? There are results and conclusion of these simulations? The simulated result significantly improves the article.

The simulation was conducted on SolidWorks. Results and conclusion have been added in section 3.

Regarding the prototype, what is the total weight, height, and length? Where is the depth sensor installed? Also, what is the feet angle in swim mode and walking? Is it a fixed value?

Thank you for pointing this out. The total weight, height, width and length of the prototype are 4.62 Kg, 390.29 mm, 232 mm, 493.88 mm. The actual design dimensions are also illustrated in the figure 5(d).

The waterproof ultrasonic sensor and water sensor are used in this robot as shown in figure 8 (a) and (b) respectively.

The feet angle in walking and swimming mode are 0° and 80°and the angle remain fixed during the feet movement.

Although Figure 3 shows the feet position, it is not clear what is the angle. Also, the figure is never referenced in the text.

The Figure 3 was referenced before on page 2. I have changed the figure. Hope it is now clear.

Reviewer 3 Report

The paper is excellently written and structured. The topic is interesting and the prototype presented, although has several possibilities of improvement, works very well and can serve as a starting point for more complex designs. However, there are several minor issues that should be addressed before being accepted for publication:

1) Affiliation and emails of all the authors (that do not appear in the manuscript).

2) References. They need to be deeply revised since there are different styles almost for every reference. Some of them are incomplete (see, for instance, [4,5,15,18]), while others have a completely different style (see, for instance, [13] and [21]; [1] and [22,23]).

3) The paragraph conformed by lines 104 to 108 has been copied (word by word) from lines 43 to 47.

4) Line 127 does not make sense. When stating that is better and more controlled, is with respect to what?

5) I miss some motion equations when describing the mechanical design of the feet, i.e., to include the equation of motion of the designed feet and the equation of motion of the system conformed by the foot together with the Klann linkage so the reader can have a more formal idea of the motion of the system.

6) Since the robot is intended to walk through different environments and since the program flow described in section 3.3.3 includes obstacle avoidance, a video showing a test in which the robot avoid an obstacle following such program flow would be appreciated.

7) Just as a curiosity, have you explore the possibility of making the robot to walk entering into a space full of water and, once the water sensor detects the water, start to swim?

Author Response

The paper is excellently written and structured. The topic is interesting and the prototype presented, although has several possibilities of improvement, works very well and can serve as a starting point for more complex designs. However, there are several minor issues that should be addressed before being accepted for publication:

1) Affiliation and emails of all the authors (that do not appear in the manuscript).

Thank you for pointing this. The original paper was different. Hence I have updated it on the revised manuscript.

2) References. They need to be deeply revised since there are different styles almost for every reference. Some of them are incomplete (see, for instance, [4,5,15,18]), while others have a completely different style (see, for instance, [13] and [21]; [1] and [22,23]).

The references have been updated.

3) The paragraph conformed by lines 104 to 108 has been copied (word by word) from lines 43 to 47.

Really sorry for that. Somehow I missed it. I have removed the paragraph.

4) Line 127 does not make sense. When stating that is better and more controlled, is with respect to what?

Thank you for pointing this out. It has been replaced. Please check line 110 to 122

5) I miss some motion equations when describing the mechanical design of the feet, i.e., to include the equation of motion of the designed feet and the equation of motion of the system conformed by the foot together with the Klann linkage so the reader can have a more formal idea of the motion of the system.

Some more information has been added. Please check section 3.1

6) Since the robot is intended to walk through different environments and since the program flow described in section 3.3.3 includes obstacle avoidance, a video showing a test in which the robot avoid an obstacle following such program flow would be appreciated.

We have recorded number of videos but I lost the hard drive and laptop during the shipment from Malaysia to Qatar.

7) Just as a curiosity, have you explore the possibility of making the robot to walk entering into a space full of water and, once the water sensor detects the water, start to swim?

I have lost the video but please check the details below

Round 2

Reviewer 2 Report

I congratulate the authors for the improvement of their paper. However, I still have some minor suggestions:

* After the sentence "During swimming in the water, four feet can go forward and backward at the same time to maximize the speed." I suggest only comment that "It has been observed during the practical tests".

* The drag force equation should be enumerated. Also, there is a problem with the rho character on it.

* Please, review caption of Figure 8 that was placed above the figure. It should be below the figure.

* On sentence "Based on motor speed, the velocity of the feet was computed as 0.03125 m/s", it is missing the footnote 1 regarding "motor speed".

* On my pdf version, the red ellipse of Figure 9 is not on the mass propriety. Please consider fixing the ellipse position or removing it.

* Each equation should be individually enumerated and referenced on the text by its number. Please adopt reference numbers instead of position references ("above" or "below") for elements on the text.

* The commentary "mass was taken from Solidworks as SHOWN in Figure 9." will better fit after acceleration equation together with "Robot will move with the acceleration of 0.01106 m/s2".

* On sentence "sensors are used in this robot as shown in figure 10(a) and (b)", please consider using uppercase letter F on "Figure".

* I would like the authors to review the diagram of Figure 14 again. It seems the button B OFF only takes effect when an obstruction is detected. Otherwise, button B is never checked. The execution flow on the negative case of condition "Obstruction detected?" goes directly to the "Go forward for 5 seconds" state. I interpreted that the robot will never stop even if the B button is OFF. It will only stop if an obstruction is detected.
